# NFE2 Truncation Mutants Protect Wild-Type NFE2 from ITCH-Dependent Degradation

**DOI:** 10.3390/ijms262412112

**Published:** 2025-12-16

**Authors:** Mirjam Elisabeth Hoeness, Franziska Zell, Titiksha Basu, Katharina Gellrich, Albert Gründer, Jana Schulze, Anja Müller, Philipp Eble, Christoph Koellerer, Anne Marie Staehle, Sarolta Bojtine Kovacs, Heike L. Pahl, Hans Felix Staehle

**Affiliations:** 1Division of Molecular Hematology, Department of Medicine I, University Medical Center Freiburg, Faculty of Medicine, University of Freiburg, 79106 Freiburg, Germany; 2Faculty of Biology, University of Freiburg, 79104 Freiburg, Germany

**Keywords:** MPN, ITCH, NFE2

## Abstract

Myeloproliferative neoplasms (MPNs) are clonal hematopoietic disorders characterized by the abnormal proliferation of myeloid cells. In addition to the main driver mutations in JAK2, MPL, and CALR, the transcription factor nuclear factor erythroid 2 (NFE2) has emerged as a key contributor to MPN pathophysiology. NFE2 expression is elevated in the majority of MPN patients, and augmented NFE2 activity in hematopoietic stem cells is sufficient to induce an MPN phenotype with spontaneous leukemic transformation in murine models. Moreover, NFE2 mutations, found in a subset of MPN patients, augment NFE2 activity and are associated with a markedly increased risk of progression to acute myeloid leukemia (AML). However, the molecular mechanism by which NFE2 mutations cause leukemogenesis is not understood. Here, we demonstrate that the E3 ubiquitin ligase ITCH mediates proteasomal degradation of wild-type (wt) NFE2 in HEK-293T cells. A gain-of-function truncation mutant, NFE2-226aa, retains the capacity to interact with ITCH but is no longer degraded. Rather, NFE2-226aa protects wt NFE2 from ITCH-dependent degradation, resulting in enhanced NFE2 activity.

## 1. Introduction

Polycythemia vera (PV), essential thrombocythemia (ET), and primary myelofibrosis (PMF) constitute a group of clonal hematologic disorders collectively termed myeloproliferative neoplasms (MPN). Mutations in the *Janus kinase 2 (JAK2)*, *calreticulin (CALR)*, and the *myeloproliferative leukemia virus oncogene (MPL)* have been identified as the main oncogenic drivers of these disorders. In addition, the transcription factor nuclear factor erythroid 2 (NFE2) plays an important role in MPN pathophysiology. NFE2 levels are increased in the majority of MPN patients, and transgenic mice expressing elevated NFE2 levels develop an MPN-like phenotype with spontaneous transformation to acute leukemia (AML), at a rate similar to that observed in MPN patients [1,2,3]. Furthermore, NFE2 mutations, found in approximately 3–9% of MPN patients, are associated with a significantly increased risk of progression to AML [4,5,6,7].

The NFE2 protein contains three key structural domains: an N-terminal transactivation domain, a central DNA-binding domain, and a C-terminal dimerization domain [8,9,10,11,12,13]. Heterodimerization with small Maf proteins is required for NFE2 to function as a transcription factor (Appendix A) [9,14]. However, the majority of NFE2 mutations identified in MPN patients do not contain a functional dimerization domain, either because of frameshift mutations that cause a premature stop, leading to a C-terminally truncated protein, or because of mutations within the dimerization domain itself.

NFE2 mutations identified in MPN patients exclusively occur heterozygously, suggesting that the presence of wild-type (wt) NFE2 is required for the mutants to exert their pathological activity. We have shown that mutant NFE2 proteins, while inactive on their own, enhance activity of the remaining wt protein, thereby promoting leukemic transformation [4,5]. However, the molecular mechanisms by which NFE2 mutants augment wt NFE2 activity are not understood.

It has been suggested that the E3 ubiquitin protein ligase ITCH post-translationally regulates NFE2 levels [15,16]. ITCH contains three functional domains: (i) an N-terminal C2 domain, which influences ITCH’s subcellular localization by enabling binding to membrane lipids [17]; (ii) four WW domains, tryptophan-rich motifs capable of binding proline-containing sequences which facilitate interactions with substrates and activator molecules [18]; (iii) a C-terminal HECT domain, which mediates the catalytic transfer of ubiquitin to target proteins [19].

Using an electrophoretic mobility shift assay (EMSA), Chen et al. previously demonstrated that ITCH binds NFE2 at the N-terminal PY motifs via its proline-rich WW domains [15]. Subsequently, Lee et al. showed that this interaction leads to ubiquitination of NFE2, resulting in its relocalization from the nucleus to the cytoplasm, which is associated with reduced transactivation activity [16].

Here, we tested the hypothesis that altered interactions between truncated NFE2 mutants and ITCH constitute a molecular mechanism by which the presence of mutant NFE2 results in augmented NFE2 activity.

## 2. Results

### 2.1. AlphaFold Predicitons of Wild-Type and Mutant NFE2

To assess the interaction potential of NFE2 with the E3 ubiquitin ligase ITCH, we first performed in silico AlphaFold predictions (Figure 1). In the predicted local distance difference test (pLDDT), which approximates model confidence [20,21], wt NFE2 exhibits low pLDDT values in the N-terminal transactivation domain and high pLDDT values in the C-terminal DNA-binding and leucine-zipper domains (Figure 1A). High pLDDT scores generally correspond to well-structured regions, whereas low pLDDT scores often indicate intrinsically disordered regions (IDRs). IDRs are commonly found in transcription factor domains and facilitate context-dependent interactions with other proteins [22,23,24,25].

We next tested two previously characterized gain-of-function NFE2 mutants in AlphaFold predictions: the first contains a small in-frame deletion in the leucine zipper heterodimerization domain (NFE2-Δ297-300, Figure 1B); the second lacks the C-terminal 153 amino acids, due to a frameshift mutation that results in a premature translational stop (truncated NFE2-226aa, Figure 1C). Both mutants augment wt NFE2 activity despite having lost their DNA-binding capability (Appendix A) [5].

Compared to wt NFE2, the NFE2-Δ297-300 mutant shows a similar overall pLDDT value of nearly 70%, whereas the truncated NFE2-226aa variant lacks the ordered C-terminal regions and therefore displays a lower overall pLDDT value of approximately 60% (Figure 1D–F). Because the NFE2-226aa mutant consists almost entirely of disordered regions, we hypothesized that this variant likely exhibits a substantially altered binding behavior toward other proteins.

Notably, the pLDDT values of the PY motifs, which have been proposed to mediate binding to ITCH (Figure 2) [15], were similar across all three predictions (Figure 1D–F). Based on our structural models, wt NFE2, NFE2-Δ297-300 and NFE2-226aa therefore all possess the ability to interact with ITCH.

### 2.2. ITCH Interacts with Wild-Type and Mutant NFE2

To test our hypotheses derived from the AlphaFold predictions, we next assessed whether wt NFE2 and ITCH physically interact. To this end, we overexpressed wt NFE2 and ITCH in HEK-293T cells and performed co-immunoprecipitation (Co-IP) assays. Western blot analysis showed that immunoprecipitation (IP) of HA-tagged wt NFE2 successfully co-precipitated ITCH (Figure 3A). Conversely, IP of MYC-tagged ITCH co-precipitated wt NFE2, confirming the interaction between the two proteins (Figure 3B). To assess protein interaction via a different technique, we performed a proximity ligation assay (PLA). Rolling circle amplification of the PLA probes attached to antibodies marking NFE2 and ITCH only produced a positive signal when both NFE2 and ITCH were co-transfected into HEK-293T cells. This provides evidence of their physical proximity, consistent with protein–protein interactions (Figure 3C).

Having confirmed that wt NFE2 interacts with ITCH, we next investigated whether ITCH also binds the NFE2-Δ297-300 and NFE2-226aa mutants. Co-IP and PLA analyses demonstrated that, similar to wt NFE2, both NFE2 mutants interact with ITCH (Figure 3D–I). These data show that interaction between NFE2 and ITCH is mediated via the N-terminal domain and remains intact in NFE2 C-terminal truncation or deletion mutants.

### 2.3. ITCH Mediates Proteasomal Degradation of Wild-Type NFE2

Having demonstrated physical binding of ITCH and NFE2, we next investigated whether this interaction modulates the stability of the NFE2 protein. We therefore performed cycloheximide (CHX) chase assays in HEK-293T. Cells were co-transfected with wt NFE2 along with either wt ITCH, an enzymatically inactive ITCH mutant harboring a point mutation in the catalytic domain (ITCH-C830A) [26] or an empty vector control. 24 h post-transfection, CHX was added and NFE2 levels were determined by Western blot at baseline and at defined time points following CHX treatment (Figure 4A). Co-expression of wt ITCH significantly reduced NFE2 levels compared to transfection of the empty vector control. In the presence of wt ITCH, NFE2 half-life was reduced from approximately 3.5 h to 2 h (Figure 4A–C). In contrast, co-expression of the inactive ITCH-C830A mutant did not decrease NFE2 levels; instead, NFE2 half-life was increased to 6.5 h in the absence of a functional ITCH E3 ligase (Figure 4A–C).

To determine whether ITCH-mediated degradation of NFE2 is proteasome-dependent, we again co-transfected wt NFE2 along with wt ITCH or an empty control into HEK-293T cells. Subsequently, cells were treated with either the proteasome inhibitor bortezomib or DMSO as a control. In cells co-transfected with the empty vector, bortezomib treatment did not significantly affect NFE2 protein levels (Figure 4D,E). However, in cells co-transfected with wt ITCH, bortezomib treatment effectively prevented the reduction in NFE2 levels, supporting our hypothesis that ITCH promotes proteasomal degradation of NFE2 (Figure 4D,E).

Next, we assessed the influence of ITCH on NFE2 function by assessing transcription factor activity using a beta-globin based luciferase reporter assay. We co-transfected HEK-293T cells with wt NFE2 and its binding partner MAF-G along with increasing amounts of wt ITCH. This resulted in a dose-dependent decrease in luciferase activity (Figure 4F). In contrast, when we repeated the experiment using the catalytically inactive mutant ITCH-C830A, NFE2-driven transactivation remained unchanged, even at higher ITCH-C830A levels (Figure 4G). ITCH-mediated proteasomal degradation therefore reduces wt NFE2 transcription factor activity.

### 2.4. ITCH Mediates Degradation of NFE2-Δ297-300 but Not of NFE2-226aa

To assess the effect of ITCH on the NFE2 mutants Δ297-300 and NFE2-226aa, HEK-293T cells were co-transfected with either mutant or wt NFE2, along with wt ITCH, enzymatically inactive ITCH-C830A or an empty control vector. Wt ITCH but not ITCH-C830A reduced protein levels of the NFE2-Δ297-300 mutant similar to the effect seen on wt NFE2 (Figure 5A–D). However, in contrast, the NFE2-226aa mutant was not degraded by wt ITCH. Instead, NFE2-226aa levels increased in the presence of both wt ITCH and ITCH-C830A (Figure 5E,F). Wt ITCH is therefore able to promote degradation of an NFE2 mutant, Δ297-300, which harbors merely a minor structural alteration, whereas the substantially altered and truncated NFE2-226aa mutant is resistant to ITCH-mediated degradation despite its confirmed binding to the ligase (Figure 3G–I).

### 2.5. NFE2-226aa Protects Wild-Type NFE2 from ITCH-Mediated Degradation

We have previously shown that various NFE2 mutants, despite lacking DNA-binding and transactivational activity, nonetheless increase protein levels of wt NFE2 [4,5]. We therefore hypothesized that NFE2-226aa protects wt NFE2 from ITCH-mediated degradation. To test this model, we co-transfected wt NFE2 with either NFE2-226aa or an empty vector control and, in addition, with either wt ITCH or an empty vector. As demonstrated above, wt ITCH reduced wt NFE2 protein levels in the absence of NFE2-226aa. However, in the presence of NFE2-226aa, wt NFE2 levels remained unaltered, despite the presence of wt ITCH (Figure 6A,B). These results support our hypothesis that the structurally altered truncation mutant acts as a decoy, binding and occupying the E3 ligase, thereby preventing it from tagging wt NFE2 for proteolytic degradation.

We also tested whether the protection mediated by NFE2-226aa allows transcriptional transactivation of the beta-globin luciferase reporter construct by wt NFE2 in the presence of wt ITCH. Consistent with our results above, in the absence of NFE2-226aa, wt ITCH significantly reduced the transactivation ability of wt NFE2. However, when NFE2-226aa was present, wt ITCH was no longer able to significantly reduce wt NFE2-mediated transactivation (Figure 6C). This protective effect was also evident when a gradient of increasing wt ITCH concentrations was applied (Figure 6D). Since we have shown that NFE2-226aa is not capable of transactivating the β-globin locus, the observed luciferase activity cannot be attributed to the presence of this mutant but must be derived from wt NFE2 [4,5]. Increased NFE2 activity in the presence of a truncating NFE2 mutant is thus due to stabilization of the remaining wt NFE2 protein.

Analysis of RNA sequencing data from Tan et al. [28] revealed that ITCH expression in PV patients is comparable to that in healthy controls (Appendix A). Likewise, ITCH mRNA levels in NFE2-mutated AML patients from the BeatAML cohort [29] are similar to those in non-mutated AML patients (Appendix A). However, analysis of NFE2 and ITCH protein expression within the BeatAML cohort [30] revealed a significant negative correlation (Appendix A). These data suggest that the mechanism by which NFE2-226aa prevents ITCH-mediated NFE2 degradation in HEK-293T cells may also be relevant in MPN and AML patients.

### 2.6. ITCH Alters the Subcellular Localization of Wild-Type NFE2 but Not of NFE2-226aa

Having assessed the impact of ITCH on the proteasomal degradation of NFE2, we next aimed to investigate how ITCH affects the subcellular localization of wt and truncated NFE2. To address this, HEK-293T cells were co-transfected with GFP-tagged wt NFE2 or GFP-tagged NFE2-226aa together with either an empty vector or mCherry-tagged ITCH (Figure 7). Subcellular localization of the respective proteins was then determined by live-cell fluorescence imaging. While wt NFE2 was predominantly enriched in the nucleus, truncated NFE2-226aa showed a much weaker nuclear staining (Figure 7A). In the presence of ITCH, a significantly larger proportion of wt NFE2 was located in the cytoplasm than in the absence of the E3 ligase. However, the addition of ITCH did not alter the localization of NFE2-226aa (Figure 7B,C).

These data further support our model that ITCH reduces the transactivation activity of wt NFE2 by altering its subcellular localization from the nucleus to the cytoplasm, where proteasomal degradation occurs (Figure 4, Figure 7 and Figure 8). NFE2-226aa, which retains the ability to interact with ITCH, is strongly enriched in the cytoplasm, where it acts as a decoy, binding ITCH, thereby preventing ITCH-mediated degradation of wt NFE2 (Figure 6, Figure 7 and Figure 8).

## 3. Discussion

We have previously shown that augmented NFE2 activity in hematopoietic stem cells, either through overexpression of the wt protein or through the presence of mutations that increase NFE2 levels causes a myeloproliferative phenotype with spontaneous transformation to acute leukemia in several mouse models [3,4,5]. In addition, in a large cohort of MPN patients, NFE2 mutations were associated with an 8-fold increased odds ratio of leukemic transformation [6]. Our laboratory has proposed a classification system that distinguishes between Type I (gain-of-function) and Type II (loss-of-function) NFE2 mutations, which are further divided into DNA-binding (Type A) and non-DNA-binding (Type B) mutations (Appendix A) [5]. We further proposed a model in which both abnormally increased and unphysiologically reduced NFE2 activity can promote leukemogenesis [5]. Therefore, we expect several distinct molecular mechanisms to contribute to leukemic transformation in the different classes of NFE2 mutants.

The various NFE2 mutations impact protein structure in different ways, affecting post-translational modifications and altering interactions with binding partners, ultimately affecting the ability to promote transcription. For only one of the various NFE2 mutations found in MPN and AML patients, have we to date been able to determine a molecular mechanism of action. Wt NFE2 is physiologically sumoylated at lysine 368 (K368), a modification proposed to interfere with activating post-translational modifications and to promote sequestration of NFE2 into promyelocytic leukemia (PML) nuclear bodies [31,32]. The Type IA NFE2-K368X mutation, identified in a patient diagnosed with triple negative ET, disrupts this sumoylation site, resulting in unphysiologically elevated transcriptional activity [32]. In this patient, we propose that augmented NFE2 activity through a gain-of-function, DNA-binding mutation drives thrombocytosis in absence of a classical MPN driver mutation.

The mechanism of action by which truncating NFE2 mutations, which lack all activity on their own, augment wt NFE2 activity has not been elucidated. Interaction with ITCH has been proposed as an additional mechanism of regulating NFE2 activity at the post-translational level. This interaction was first described by Chen et al., who demonstrated that ITCH binds NFE2 (Figure 2) [15]. Subsequently Lee et al. proposed that ITCH diminishes the transactivating capacity of NFE2 by altering its subcellular localization relocating the transcription factor into the cytoplasm [16].

Even though we also observed a redistribution of NFE2 from the nucleus to the cytoplasm in the presence of ITCH, our data contradict the model of altered intracellular localization as the sole mechanism of ITCH-mediated regulation of NFE2 transcriptional activity. While we confirm binding of NFE2 to ITCH, we demonstrate that this interaction not only promotes relocalization but also induces proteasomal NFE2 degradation, resulting in reduced transactivation activity (Figure 4 and Figure 7). Notably, presence of the NFE2-226aa mutant found in MPN and AML patients inhibits ITCH-mediated degradation of wt NFE2. Our in vitro data assessed in HEK-293T cells provide the first molecular mechanism by which an NFE2 truncation mutant, which has no activity on its own, enhances wt NFE2 activity. We propose that the structurally altered, truncated NFE2 protein binds and engages the E3 ligase, acting as a decoy, delaying or preventing ITCH-mediated degradation (Figure 6).

Lee et al., have shown that wt NFE2 contains six lysines, K108, K137, K215, K234, K241, and K368, which are targeted for polyubiquitination, leading to subsequent degradation [33]. Moreover, phosphorylation at serines S157 and S346 enhances NFE2 ubiquitination. Truncated NFE2-226aa has lost the three major C-terminal ubiquitination sites, K234, K241, and K368, as well as the S346 phosphorylation site. It therefore appears likely that NFE2-226aa undergoes less efficient polyubiquitination, which could explain its resistance to ITCH-mediated degradation.

In contrast to the NFE2-226aa truncation mutant, the NFE2-Δ297-300 mutant remains susceptible to ITCH-dependent degradation (Figure 5). Previous work from our laboratory demonstrated that the NFE2-Δ297-300 mutant belongs to the mutants with the highest potential to enhance wt NFE2 despite lacking intrinsic transactivation capability [4]. Our findings suggest that its mechanism of action is unlikely to involve an altered interaction with ITCH (Figure 8). Instead, it is more likely that each NFE2 mutant operates through a distinct pathophysiological mechanism. Further investigation is therefore needed to elucidate the precise molecular pathways through which NFE2-Δ297-300 and other NFE2 mutants exert their effects. A deeper understanding of these mechanisms may ultimately contribute to the development of novel therapeutic strategies for normalizing NFE2 activity in MPN patients. Because of the significant risk conferred by the presence of NFE2 mutations, normalizing NFE2 activity may constitute a clinically meaningful intervention in this patient population.

## 4. Materials and Methods

### 4.1. HEK-293T Cells

HEK-293T cells were obtained and cultured following the recommendations from DMSZ (German Collection of Microorganisms and Cell Cultures). The cells were maintained at 37 °C in a humidified atmosphere containing 5% CO_2_ in DMEM (Gibco-Thermo Fisher Scientific, 41966-029, Waltham, MA, USA), supplemented with 10% fetal calf serum (Gibco-Thermo Fisher Scientific, 10270-106), 1% penicillin/streptomycin (Lonza, 17-602E, Basel, Switzerland), and 1% L-glutamine (Gibco-Thermo Fisher Scientific, 25030). Cells were cultured in a Heracell 240i CO_2_ incubator (Thermo Fisher Scientific, 51032875). Upon reaching > 90% confluency, cells were detached from the culture flask using trypsin (Gibco-Thermo Fisher Scientific, 12605010) and subcultured at a 1:10 ratio.

### 4.2. Transient Transfection

Transient transfection was performed using either the calcium chloride (CaCl_2_) or the polyethylenimine (PEI) method. CaCl_2_-mediated transfection was conducted 4–6 h after seeding of HEK-293T cells, whereas PEI-mediated transfection was carried out 20–24 h post-seeding. For CaCl_2_ transfection, plasmid DNA was mixed with double-distilled water (ddH_2_O), 2.5 M CaCl_2_ (Sigma-Aldrich–Merck, 10043-52-4, Darmstadt, Germany), and BES buffer, then added dropwise to the cells (Appendix A). The BES buffer contained 50 mM N,N-bis(2-hydroxyethyl)-2-aminoethanesulfonic acid (Sigma-Aldrich–Merck, B2891-25G), 280 mM NaCl (VWR, 27810.295, Radnor, PA, USA), and 1 mM Na_2_HPO_4_·2H_2_O (Roth, 4984.1, Karlsruhe, Germany), adjusted to pH 6.9 with HCl (VWR, 20257.296). For PEI transfection, plasmid DNA was mixed with DMEM and PEI (Polysciences, Warrington, PA, USA, 23966-1) before being added to the cells (Appendix A). A list of all plasmids used in this study is provided in Appendix A.

### 4.3. Protein Isolation

Proteins were isolated 24–48 h after transfection using the RIPA lysis protocol. The RIPA buffer consisted of 150 mM NaCl (VWR, 278102-95), 1% NP-40 (Roche, 11332473001, Basel, Switzerland), 0.1% sodium deoxycholate (Sigma-Aldrich–Merck, D-6750), and 50 mM Tris-HCl, pH 8.0 (Roth, 4855.3). Shortly before cell lysis, 1% EDTA-free Complete Protease Inhibitor Cocktail (Roche, 04693116001), as well as Phosphatase Inhibitor Cocktails 2 and 3 (Sigma-Aldrich–Merck, P5726 or P0044, respectively), were added from 100× stock solutions. HEK-293T cells were detached from the surface by pipetting and washed with ice-cold PBS. After centrifugation at 300 g for 5 min at 4 °C, the cell pellet was resuspended in 80–120 µL RIPA buffer per one million cells. The lysate was incubated on ice for 20 min with occasional vortexing, followed by centrifugation at 13,000 rpm for 10 min at 4 °C in a 5424R centrifuge (Eppendorf). The resulting protein-containing supernatant was transferred to a new microcentrifuge tube. Protein concentrations were determined using the Lowry assay (Bio-Rad, 5000112, Hercules, CA, USA), and samples were stored at −80 °C until further use.

### 4.4. Co-Immunoprecipitation (Co-IP)

Co-IPs were performed to test the interaction between ITCH and wt NFE2, NFE2-226aa, or NFE2-Δ297-300. 3 × 10^6^ HEK-293T cells were seeded in 10 cm dishes (Sarstedt, Nümbrecht, Germany, 833902). 4–6 h after seeding, cells were co-transfected using the CaCl_2_ method. Cells were harvested 36 h post-transfection, and proteins were isolated using the RIPA lysis buffer protocol. 10% of the protein lysate was retained as input. Co-IPs were performed on the remaining lysate using either the Pierce™ HA-Tag Magnetic IP/Co-IP Kit (Thermofisher Scientific, 88838) or the Pierce™ Myc-Tag Magnetic IP/Co-IP Kit (Thermofisher Scientific, 23620), according to the manufacturer’s instructions.

### 4.5. Immunoblotting

Sodium dodecyl sulfate–polyacrylamide gel electrophoresis (SDS-PAGE) and Western blotting were performed as previously described to assess protein expression. [2,34] Immune complexes were detected using chemiluminescence (PerkinElmer, NEL103001EA, Waltham, MA, USA). Densitometric analysis was conducted using the ImageJ software version 1.53m (Fiji distribution) (National Institutes of Health). [35] The antibodies used for immunoblotting are listed in Appendix A.

### 4.6. Proximity Ligation Assay (PLA)

The PLA was used to visualize the interaction between ITCH and wt NFE2, NFE2-226aa, or NFE2-Δ297-300. 0.5 × 10^6^ HEK-293T cells were seeded per well in 6-well plates (Sarstedt, 83.3920, Nümbrecht, Germany). 24 h after seeding, cells were co-transfected using the PEI method. After an additional 24 h, cells were detached from the plates and 0.1 × 10^6^ transfected cells were re-seeded into 8-well chamber slides (ibidi GmbH, 80826, Gräfelfing, Germany). 24 h after re-seeding, cells were fixed with 2% formalin (SAV Liquid Production GmbH, FN-5000-4-1, Flintsbach, Germany) and permeabilized with 0.2% Triton X-100 (Sigma-Aldrich, T8787, St. Louis, MO, USA). The PLA was then performed using the Duolink^®^ In Situ Red Starter Kit (Merck, DUO92101, Darmstadt, Germany), according to the manufacturer’s instructions. Cells were imaged using a Zeiss LSM880 confocal microscope (Carl Zeiss AG, Oberkochen, Germany) with the ZEN 2.3 SP1 (Carl Zeiss AG, Oberkochen, Germany) software.

### 4.7. Cycloheximide (CHX) Chase

The CHX chase assay was employed to assess the protein stability of wt NFE2 in the presence of ITCH. 0.5 × 10^6^ HEK-293T cells were seeded per well in 6-well plates (Sarstedt, 833920). 4–6 h after seeding, cells were co-transfected using the CaCl_2_ method. 24 h post-transfection, CHX (Sigma-Aldrich-Merck, C4859, St. Louis, MO, USA) was added to each well at a final amount of 1 µg. Cells were harvested at 0, 1, 2, 3, 4, and 5 h after CHX addition, flash-frozen in liquid nitrogen, and stored at −80 °C until protein extraction using RIPA buffer.

### 4.8. Bortezomib Assay

The bortezomib treatment assay was conducted to determine whether proteasome inhibition affects the degradation of NFE2 in the presence of ITCH. 0.5 × 10^6^ HEK-293T cells were seeded per well in 6-well plates. 24 h after seeding, cells were co-transfected using the PEI method. 24 h post-transfection, bortezomib (Merck, Darmstadt, Germany 179324-69-7) was added to a final concentration of 100 nM. Cells were harvested 4 h after treatment, and proteins were extracted using the RIPA lysis buffer protocol.

### 4.9. Luciferase Assay

Luciferase assays were performed as previously described to quantify NFE2 activity using a β-globin-derived reporter construct containing the NFE2 binding site [32,34]. β-globin is one of the best-characterized NFE2 target genes and therefore represents an established and widely accepted readout for NFE2 activity [4,14,32,36,37,38]. 2 × 10^5^ HEK-293T cells were seeded per well in 12-well plates (Greiner Bio-One, Kremsmünster, Austria 665180). 24 h post-seeding, cells were co-transfected using the PEI method with an HBB promoter-driven firefly luciferase reporter construct, a Renilla luciferase control plasmid, and expression vectors encoding wt NFE2, MAF-G, wt ITCH, ITCH-C830A, or NFE2-226aa, depending on the experimental condition (Appendix A). Cells were harvested 12 h after transfection, and luciferase activity was measured using the Dual-Luciferase^®^ Reporter Assay System (Promega Gmbh-Promega Corporation, Madison, WI, USA, E1960) in a Microplate Luminometer LB 96 V (Berthold EG&G-Berthold Technologies, Zug, Switzerland 23300). Luciferase signals were detected using an Infinite 200 microplate reader (Tecan AG, Männerdorf, Switzerland, 30190087). Firefly luciferase activity was normalized to Renilla luciferase activity to compensate for variability in transfection efficiency.

### 4.10. Live-Cell Fluoresence Imaging

Live-cell fluorescence imaging was used to visualize the subcellular localization of wt NFE2, NFE2-226aa, and ITCH. A total of 1 × 10^4^ HEK-293T cells were seeded per well in µ-Slide ibiTreat 8-well plates (ibidi, 80806, Gräfelfing, Germany). 24 h after seeding, cells were co-transfected with different combinations of wt GFP-NFE2, GFP-NFE2-226aa, mCherry-ITCH, and empty vector using the PEI method. After an additional 24 h, cells were incubated for 1 h with SPY650 (Spirochrome, SC501, Stein am Rhein, Switzerland), diluted 1:100 in DMEM, for nuclear staining. The staining solution was then replaced with culture medium, and live imaging was performed at 37 °C and 5% CO_2_ using a Zeiss LSM880 confocal microscope (Carl Zeiss AG, Oberkochen, Germany) with the ZEN 2.3 SP1 software.

### 4.11. Analysis of RNA Sequencing Data

For PV patients, RNA-seq data from Tan et al. [28] were accessed via the accession number GSE145802. For AML patients, raw RNA-seq counts, WES data, and clinical summaries were downloaded from the official BEAT2 repository (https://biodev.github.io/BeatAML2/) accessed on 6 December 2025 [29]. Counts were prefiltered for genes with meaningful expression values and then variance-stabilized using the DESeq2 vst() function [39].

### 4.12. Analysis of Mass Spectometry Data

Mass spectrometry data from AML patients within the BeatAML cohort were accessed from Pino et al. [30]. Transformed and normalized intensities, as provided by the original studies, were used without further processing.

### 4.13. Statistical Analysis

Unpaired Student’s *t*-tests were used to determine whether a significant (*p* < 0.05) difference existed between two groups. Analyses were performed using the GraphPad Prism 10 software (GraphPad Software, Boston, MA, USA).

### 4.14. AlphaFold Structural Predictions

AlphaFold structure predictions were generated using AlphaFold 2.0. Visualization of the predicted structures was performed using the following code: DOI 10.5281/zenodo.6548465.

## Figures and Tables

**Figure 1 ijms-26-12112-f001:**
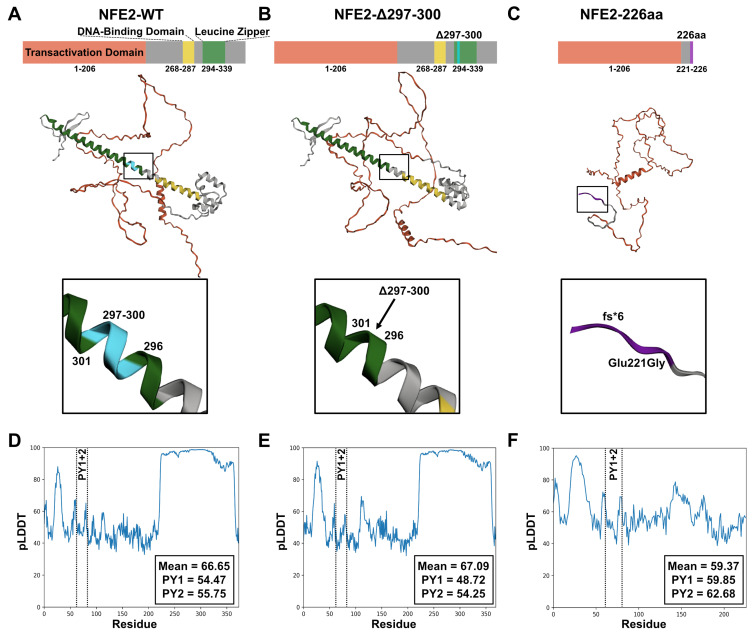
Wild-type NFE2 and its Mutant Variants NFE2-Δ297-300 and NFE2-226aa. (**A**–**C**) Protein structures of (**A**) NFE2-WT, (**B**) NFE2-Δ297-300, and (**C**) NFE2-226aa. Top: Schematic representation of the main domains. Middle: AlphaFold structural prediction. Bottom: Close-up view of the mutated regions. Color coding: red, transactivation domain; yellow, DNA-binding domain; green, leucine zipper domain; turquoise, location of the Δ297-300 deletion; purple, C-terminal frameshift in the 226aa mutant. (**D**–**F**) Predicted local distance difference test (pLDDT) scores of (**A**) NFE2-WT, (**B**) NFE2-Δ297-300, and (**C**) NFE2-226aa. The box displays the overall mean pLDDT score, as well as the individual scores for the PY1 and PY2 motifs.

**Figure 2 ijms-26-12112-f002:**
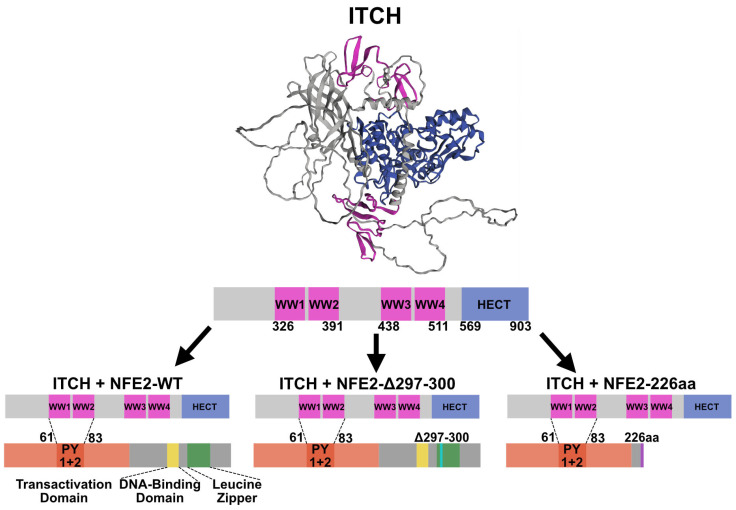
Interaction between ITCH and NFE2. (Top): AlphaFold structural prediction of ITCH. Predicted Local Distance Difference Test (pLDDT) values are provided in Appendix A. (Middle): Schematic representation of the main functional domains of ITCH. (Bottom): Proposed interaction between the WW1+2 domains of ITCH and the PY1+2 regions of NFE2 (dashed line), based on Lee et al. and Chen et al. [15,16]. Color coding: pink, WW domains; blue, HECT domain; red, transactivation domain; yellow, DNA-binding domain; green, leucine zipper domain; turquoise, position of the Δ297-300 deletion; purple, C-terminal frameshift in the 226aa mutant.

**Figure 3 ijms-26-12112-f003:**
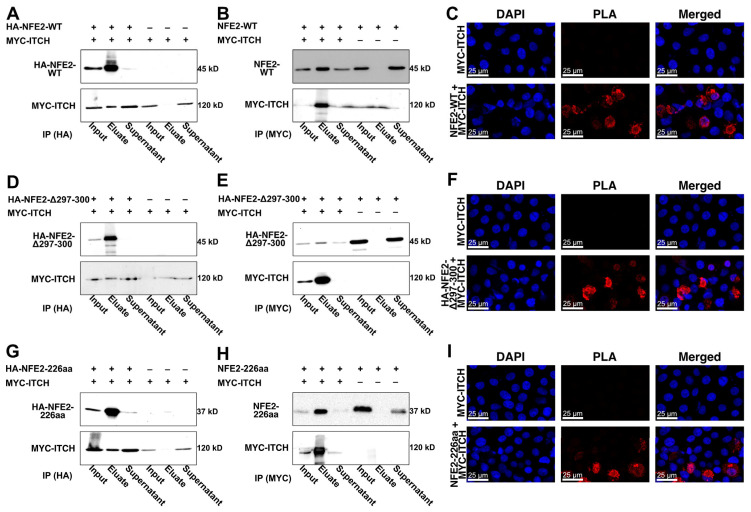
ITCH Interacts with Wild-type and Mutated NFE2. (**A**–**I**) HEK-293T cells were co-transfected with MYC-tagged ITCH and plasmids coding for (**A**–**C**) NFE2-WT, (**D**–**F**) NFE2-Δ297-300, and (**G**–**I**) NFE2-226aa. (**A**,**B**,**D**,**E**,**G**,**H**) Co-IPs followed by Western blot analysis. Co-IPs were performed using either (**A**,**D**,**G**) anti-HA or (**B**,**E**,**H**) anti-MYC antibodies. For each condition, input, eluate, and supernatant fractions were analyzed as indicated. Blots were probed with antibodies against NFE2 (up) or MYC (bottom). (**C**,**F**,**I**) Fluorescence images from PLAs in cells transfected with MYC-ITCH alone (upper panels) or MYC-ITCH together with NFE2 constructs (lower panels). DAPI nuclear staining (left), PLA signal (middle), and merged images (right) are shown.

**Figure 4 ijms-26-12112-f004:**
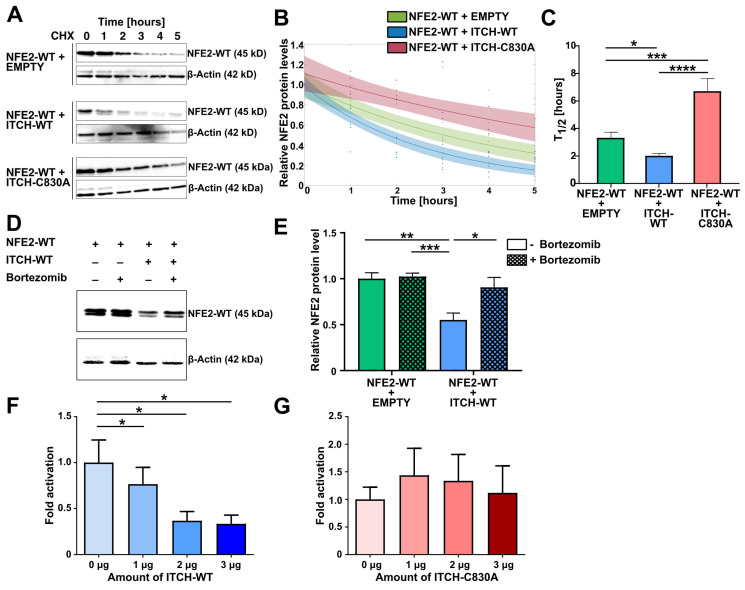
ITCH Mediates Proteasomal Degradation of Wild-type NFE2. (**A**–**C**) CHX chase in HEK-293T cells. Cells were harvested at baseline and at 1, 2, 3, 4, and 5 h following CHX treatment. (**A**) Representative Western blot images of cells co-transfected with NFE2-WT and either empty vector (top), ITCH-WT (middle), or ITCH-C830A (bottom). Blots were probed with an anti-NFE2 antibody. β-actin was used as a loading control. (**B**,**C**) Densitometric analysis was performed to determine (**B**) the decay rate and (**C**) the half-life of NFE2 co-transfected with empty vector (green), ITCH-WT (blue), or ITCH-C830A (red). *n* = 3–8 per condition. (**B**) The relative NFE2 expression at each time point compared to baseline (0 h) was fitted to an exponential function: f(x) = b⋅e^−ax^, where x is time (in hours), b is the initial protein level at time zero, and a is the decay rate. Decay rates with 95% confidence intervals: a (empty vector) = 0.2277 (0.1581, 0.2972); a (ITCH-WT) = 0.3188 (0.2317, 0.4059); a (ITCH-C830A) = 0.1316 (0.0651, 0.1981). Confidence intervals are shown as semi-transparent shaded areas matching the color of each respective fit. Non-overlapping confidence bands indicate a significant difference. (**C**) Half-lives were calculated using the equation T_1/2_ = ln(2)/a. For each data point, decay rates were determined from the exponential decay model f(x) = e^−ax^, with signal intensities normalized to baseline expression at the respective time point. (**D**,**E**) Bortezomib treatment assay in HEK-293T cells co-transfected with NFE2-WT and either empty vector or ITCH-WT. (**D**) Representative Western blot images. Blots were probed with an anti-NFE2 antibody. β-actin was used as a loading control. (**E**) Relative NFE2 protein levels were determined by densitometric analysis as described by Rees et al. [27] *n* = 5 per condition. (**F**,**G**) Luciferase reporter vectors were co-transfected into HEK-293T cells together with expression plasmids encoding NFE2, MAF-G, and increasing concentrations of (**F**) ITCH-WT or (**G**) ITCH-C830A. Data were normalized to co-transfection with 0 µg ITCH, which was set to 1. *n* = 4–5 per condition. (**C**,**E**–**G**) Graphs represent mean ± SEM. * *p* < 0.05, ** *p* < 0.01, *** *p* < 0.001, **** *p* < 0.0001 by Student’s *t*-tests. (**A**,**D**) NFE2-WT appears as a double band in Western blots [8,10,11].

**Figure 5 ijms-26-12112-f005:**
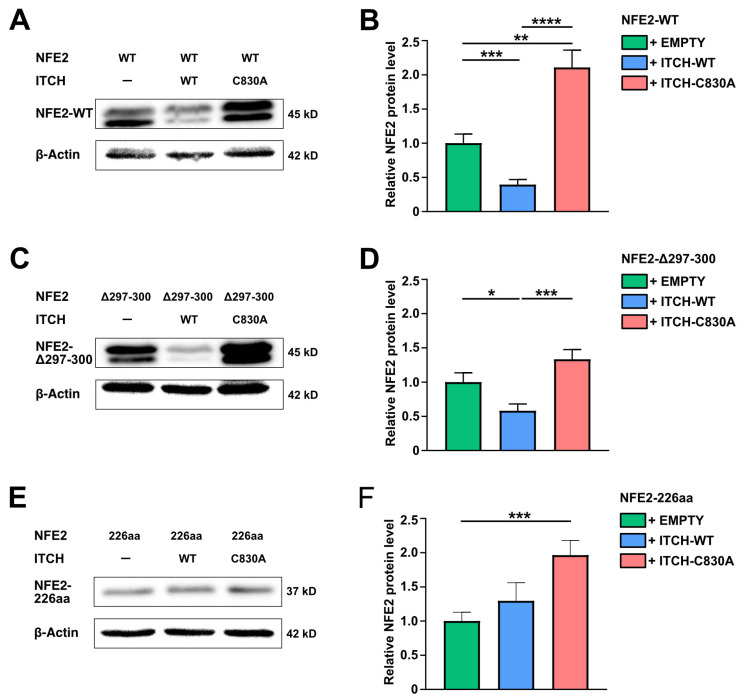
ITCH Mediates Degradation of NFE2-Δ297-300 but not of NFE2-226aa. (**A**–**F**) HEK-293T cells were co-transfected with (**A**,**B**) NFE2-WT, (**C**,**D**) NFE2-Δ297-300, or (**E**,**F**) NFE2-226aa along with empty vector, ITCH-WT, or ITCH-C830A, as indicated. (**A**,**C**,**E**) Representative Western blot images. Blots were probed with an anti-NFE2 antibody. NFE2-WT [8,10,11] and NFE2-Δ297-300 appears as a double band in Western blots. β-actin was used as a loading control. (**B**,**D**,**F**) Relative NFE2 expression was determined by densitometric analysis as described by Rees et al. [27] *n* = 9–12 per condition. Graphs represent mean ± SEM. * *p* < 0.05, ** *p* < 0.01, *** *p* < 0.001, **** *p* < 0.0001 by Student’s *t*-tests.

**Figure 6 ijms-26-12112-f006:**
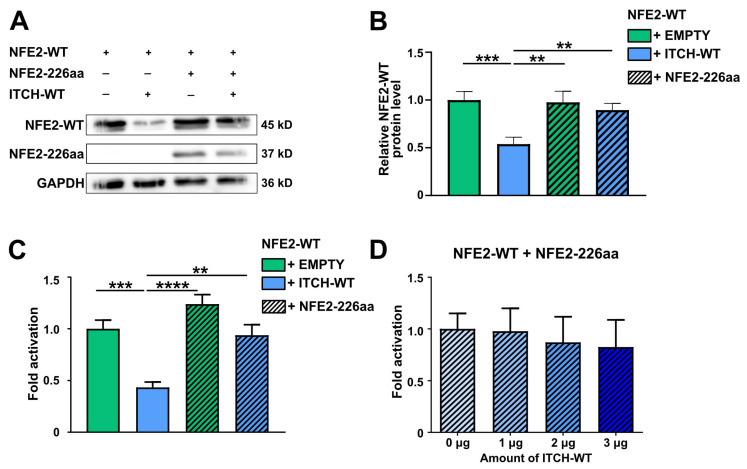
NFE2-226aa Protects Wild-type NFE2 from ITCH-mediated Degradation. (**A**,**B**) HEK-293T cells were co-transfected with NFE2-WT along with ITCH-WT and NFE2-226aa, as indicated. (**A**) Representative Western blot image. The blot was probed with an anti-NFE2 antibody. NFE2-WT appears as a double band in Western blots. [8,10,11] GAPDH was used as a loading control. (**B**) Relative NFE2 expression was determined by densitometric analysis as described by Rees et al. [27]. *n* = 9–11 per condition. (**C,D**) Luciferase reporter vectors were co-transfected into HEK-293T cells together with expression plasmids encoding NFE2, MAF-G, and (**C**) varying combinations of ITCH-WT and NFE2-226aa, as indicated, or (**D**) NFE2-226aa as well as increasing concentrations of ITCH-WT. Data were normalized to (**C**) NFE2-WT without ITCH-WT or (**D**) co-transfection with 0 µg ITCH-WT. *n* = 4–5 per condition. (**B**–**D**) Graphs represent mean ± SEM. ** *p* < 0.01, *** *p* < 0.001, **** *p* < 0.0001 by Student’s *t*-tests.

**Figure 7 ijms-26-12112-f007:**
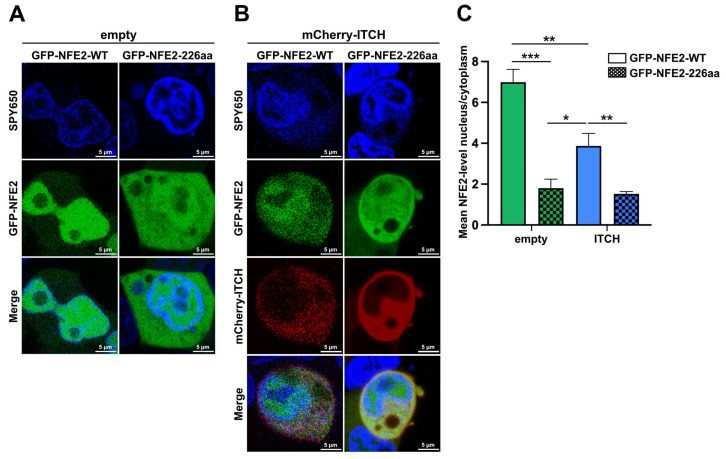
Subcellular Localization of NFE2 is Altered by C-terminal Truncation and Presence of ITCH. (**A**–**C**) HEK-293T cells were co-transfected with either (**A**) empty vector or (**B**) mCherry–ITCH together with GFP–NFE2-WT (left) or GFP–NFE2-226aa (right). The growth medium was supplemented with SPY650 for live-cell nuclear staining. (**A,B**) Representative fluorescence images showing SPY650 (blue), GFP–NFE2 (green), mCherry–ITCH (red), or merged signals as indicated. (**C**) Quantification of the nuclear-to-cytoplasmic NFE2 signal ratio. Data represent mean ± SEM. *n* = 5 cells per condition. Graphs represent mean ± SEM. * *p* < 0.05, ** *p* < 0.01, *** *p* < 0.001 by Student’s *t*-tests.

**Figure 8 ijms-26-12112-f008:**
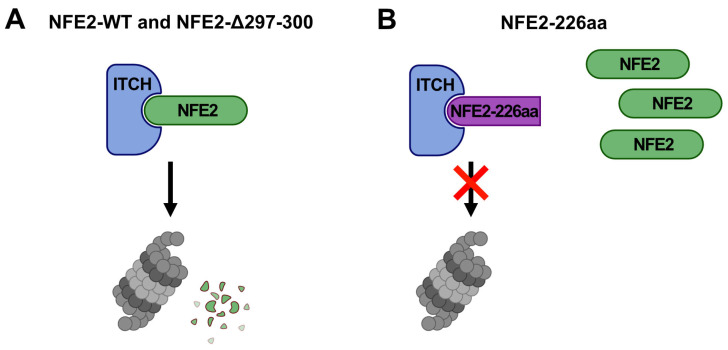
Proposed Mechanism of the Interaction between ITCH and NFE2. (**A**) Interaction with ITCH leads to proteasomal degradation of both NFE2-WT and NFE2-Δ297-300. (**B**) Although NFE2-226aa retains the ability to interact with ITCH, it is resistant to ITCH-mediated degradation. Furthermore, the presence of NFE2-226aa prevents ITCH-mediated degradation of NFE2-WT. Panels created with BioRender.com (https://BioRender.com/sbexgff, accessed on 10 November 2025).

## Data Availability

The manuscript does not include any code or large sequencing datasets. Requests for original images or plasmids should be directed to the corresponding author.

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
