# Peer review of "NFE2 Truncation Mutants Protect Wild-Type NFE2 from ITCH-Dependent Degradation"

_ijms, 2025, doi:10.3390/ijms262412112_

Round 1

Reviewer 1 Report

Comments and Suggestions for Authors

This manuscript provides clear mechanistic insight into how the truncation mutant NFE2-226aa protein acts as a decoy for E3 ubiquitin ligase ITCH, stabilizing wild-type NFE2 and explaining the gain-of-function phenotype despite lacking transcriptional activity. The study is supported by a robust experimental design, employing multiple complementary approaches including co-IP, PLA, CHX chase, bortezomib rescue, and luciferase reporter assays. Overall, the manuscript is logically structured and nicely written with clear data presentation.

The major limitation of the manuscript lies in its exclusive reliance on HEK-293T cells, which are non-hematopoietic and may not fully recapitulate NFE2 biology, highlighting the need for validation in more disease-relevant models.

Major points

  1. All experiments regarding NFE2 and ITCH interaction are performed in HEK-293T cells. Do Authors possess any evidence that these mechanisms are relevant in a disease-specific context? For example, have the Authors measured ITCH levels in NFE2 mutant vs. wt MPN patient samples? If not, this limitation of the study should be clearly stated in the Discussion section, and Abstract should be re-written.
  2. The Authors could perform ubiquitination assays to prove that mutant NFE2-226aa is not ubiquitinated. This experiment would add a missing piece to the proposed molecular mechanism. Based on the data presented, the NFE2-226aa interacts with ITCH, but it is not degraded, resulting in a non-productive complex that prevents degradation of wt NFE2. Truncation mutation likely affects regions required for ubiquitin transfer.

Minor points

  1. The Authors state they used PCR amplification for PLA, but show immunofluorescence images on Figure 2C, F, and I. Did the Authors mean ”rolling circle amplification of the PLA probes” instead of “PCR amplification of the primers” (Line 87)? Please clarify.
  2. The Authors should justify the choice of β-globin reporter for NFE2 transactivation. Choosing a disease-related NFE2 target gene would strengthen the physiological relevance in the context of MPN.
  3. In the manuscript, Alpha Fold predicted structures of wt and mutant NFE2 and their interaction with ITCH are used too descriptively and placed in the Introduction and Discussion sections rather than Results. The structural analysis could be significantly strengthened by providing numerical values, interpretations and explanations of the AlphaFold predictions. The Authors could extract a possible explanation for how mutant NFE2-226aaa blocks ITCH (does it compete with wt NFE2 for binding, does it block enzyme activity, or both).

Author Response

Major points

Comment 1: All experiments regarding NFE2 and ITCH interaction are performed in HEK-293T cells.      Do Authors possess any evidence that these mechanisms are relevant in a disease-specific context?      For example, have the Authors measured ITCH levels in NFE2 mutant vs. wt MPN patient samples?      If not, this limitation of the study should be clearly stated in the Discussion section, and Abstract should be re-written.

Response 1: We agree with the reviewer’s point and have now clarified in the abstract and discussion that the experiments were performed in HEK-293T cells.      Primary material from NFE2-mutated MPN patients is very rare and currently not available in our laboratory.      To further address this point we therefore analyzed publicly available datasets to characterize ITCH levels in MPN and AML patients.

Analysis of RNA-seq data from Tan et al. [1] revealed that patients with polycythemia vera display ITCH mRNA levels comparable to healthy controls (Suppl.      Fig. 3A).      The presence of ITCH in MPN patients is necessary for the proposed mechanism of ITCH-mediated NFE2 degradation.

We also analyzed RNA-seq data from the BeatAML cohort [2], which includes six patients with NFE2-truncated variants, and found that these patients have ITCH levels comparable to AML patients without NFE2 mutations (Suppl.      Fig. 3B–C).      The presence of ITCH in NFE2-mutated patients is necessary for the proposed mechanism in which C-terminally deleted NFE2 variants protect wild-type NFE2 by acting as a decoy for ITCH.

Although mRNA expression may reflect protein levels, direct measurement of protein abundance offers more definitive information.      For a subset of patients in the BeatAML cohort, mass spectrometry data are available.      [3] Notably, this dataset revealed a significant negative correlation between NFE2 and ITCH protein levels (Suppl.      Fig. 3D).      Collectively, these data suggest that the ITCH-mediated degradation of NFE2 observed in HEK-293T cells may also be relevant in MPN and AML patients.

Comment 2: The Authors could perform ubiquitination assays to prove that mutant NFE2-226aa is not ubiquitinated. This experiment would add a missing piece to the proposed molecular mechanism. Based on the data presented, the NFE2-226aa interacts with ITCH, but it is not degraded, resulting in a non-productive complex that prevents degradation of wt NFE2. Truncation mutation likely affects regions required for ubiquitin transfer.

Response 2: In this comment, the reviewer raises a very interesting point regarding a potential mechanism of the NFE2-226aa mutant. According to Lee et al., wt NFE2 contains six ubiquitination sites, K108, K137, K215, K234, K241, and K368, which are targeted for polyubiquitination, leading to subsequent degradation. Among these, the two N-terminal sites, K108 and K137, have been described as less efficient compared to the other four sites. [4] Moreover, Lee et al. report that phosphorylation of wt NFE2 at S157 and S346 enhances its ubiquitination.

Truncated NFE2-226aa has lost the three major C-terminal ubiquitination sites, K234, K241, and K368, as well as the S346 phosphorylation site. Based on these data, it appears likely that NFE2-226aa undergoes less efficient polyubiquitination, which could explain its resistance to ITCH-mediated degradation. However, reduced polyubiquitination may not necessarily be the underlying mechanism. It is also possible that NFE2-226aa binds to ITCH and becomes ubiquitinated but is unable to dissociate from the E3 ligase, thereby preventing ITCH-mediated degradation of wt NFE2 even in its ubiquitinated form.

Although reduced ubiquitination might explain the increased stability of NFE2-226aa, we were not able to experimentally address this question due to the limited timeframe of the revision. We have therefore included a discussion of the ubiquitination and phosphorylation sites described by Lee et al. in the context of our data in the discussion section.

Minor points

Comment 1: The Authors state they used PCR amplification for PLA, but show immunofluorescence images on Figure 2C, F, and I. Did the Authors mean ”rolling circle amplification of the PLA probes” instead of “PCR amplification of the primers” (Line 87)? Please clarify.

Response 1: We thank the reviewer for pointing this out. It is correct that microscopy-based PLA relies on rolling circle amplification (RCA) of the ligated oligonucleotide probes, not PCR amplification. We inadvertently used the wrong term in the manuscript. We have corrected the sentence accordingly.

Comment 2: The Authors should justify the choice of β-globin reporter for NFE2 transactivation. Choosing a disease-related NFE2 target gene would strengthen the physiological relevance in the context of MPN.

Response 2: We thank the reviewer for raising this important point. It is correct that β-globin itself is not described as a classical disease-driving gene in MPN. However, β-globin is central to erythropoiesis and therefore contributes critically to the phenotype of patients with polycythemia vera, who display primary erythrocytosis. Moreover, β-globin is one of the best-characterized NFE2 target genes [5-7], and in NFE2-driven MPN mouse models, we previously demonstrated an association between the MPN phenotype and increased β-globin expression. [8,9] The β-globin-derived reporter construct containing the NFE2 binding site from the β-globin locus is an established and widely accepted readout of NFE2 activity in luciferase assays, and it has been used extensively to assess NFE2-dependent transcription. [9-11] We have also included parts of this explanation in the materials and methods section, and hope that the underlying literature is sufficient to justify the choice of the β-globin reporter construct.

Comment 3:In the manuscript, Alpha Fold predicted structures of wt and mutant NFE2 and their interaction with ITCH are used too descriptively and placed in the Introduction and Discussion sections rather than Results.

Response 3: In response to this reviewer’s comment, we have now removed the AlphaFold-predicted structures from the introduction and discussion and included a dedicated section within the results to present these analyses (see results section 2.1 AlphaFold predictions of Wild-Type and Mutant NFE2).

The structural analysis could be significantly strengthened by providing numerical values, interpretations and explanations of the AlphaFold predictions. The Authors could extract a possible explanation for how mutant NFE2-226aaa blocks ITCH (does it compete with wt NFE2 for binding, does it block enzyme activity, or both).

We thank the reviewer for pointing out that the AlphaFold analyses can be strengthened by providing numerical values and explanations. In the revised Figure 1, we have now included predicted local distance difference test (pLDDT) scores for wt NFE2, NFE2-Δ297–300, and NFE2-226aa, and used these values to provide a possible explanation for how the NFE2-226aa mutant blocks ITCH (see revised Fig. 1D-F). pLDDT scores indicate the confidence of an AlphaFold prediction. High pLDDT values
correspond to well-structured regions, whereas low pLDDT values often correlate with disordered regions, which are more difficult to predict reliably. [12-15] Based on these scores, we extracted the following hypotheses:

(i) The N-terminal transactivation domain of wt NFE2 shows relatively low pLDDT values compared to the structured DNA-binding and leucine zipper domains at the C-terminal end. NFE2-Δ297–300 has an overall pLDDT of nearly 70%, similar to wt NFE2, whereas NFE2-226aa lacks the ordered C-terminal regions and therefore displays a lower overall pLDDT of approximately 60% (Fig. 1D-F). Because NFE2-226aa consists almost entirely of disordered regions, we hypothesized that it likely exhibits an altered binding behavior toward other proteins.

(ii) Despite its overall low pLDDT, NFE2-226aa retains high-confidence regions corresponding to PY motifs, which have been proposed to mediate binding to ITCH. [16] Based on this observation, we hypothesize that NFE2-226aa still binds to ITCH, a hypothesis confirmed by our in vitro experiments (Fig. 2).

Together, these analyses support the hypothesis that truncated NFE2-226aa still binds to ITCH and that its disordered regions cause it to adopt a conformation that may act as a decoy to block ITCH function.

References
1. Tan, G.; Wolski, W.E.; Kummer, S.; Hofstetter, M.; Theocharides, A.P.A.; Manz, M.G.; Aebersold, R.; Meier-Abt, F. Proteomic identification of proliferation and progression markers in human polycythemia vera stem and progenitor cells. Blood Adv 2022, 6, 3480-3493, doi:10.1182/bloodadvances.2021005344.
2. Bottomly, D.; Long, N.; Schultz, A.R.; Kurtz, S.E.; Tognon, C.E.; Johnson, K.; Abel, M.; Agarwal, A.; Avaylon, S.; Benton, E.; et al. Integrative analysis of drug response and clinical outcome in acute myeloid leukemia. Cancer Cell 2022, 40, 850-864 e859, doi:10.1016/j.ccell.2022.07.002.
3. Pino, J.C.; Posso, C.; Joshi, S.K.; Nestor, M.; Moon, J.; Hansen, J.R.; Hutchinson- Bunch, C.; Gritsenko, M.A.; Weitz, K.K.; Watanabe-Smith, K.; et al. Mapping the proteogenomic landscape enables prediction of drug response in acute myeloid leukemia. Cell Rep Med 2024, 5, 101359, doi:10.1016/j.xcrm.2023.101359.
4. Lee, T.L.; Shyu, Y.C.; Hsu, P.H.; Chang, C.W.; Wen, S.C.; Hsiao, W.Y.; Tsai, M.D.; Shen, C.K. JNK-mediated turnover and stabilization of the transcription factor p45/NF-E2 during differentiation of murine erythroleukemia cells. Proc Natl Acad Sci U S A 2010, 107, 52-57, doi:10.1073/pnas.0909153107.
5. Blank, V.; Kim, M.J.; Andrews, N.C. Human MafG is a functional partner for p45 NF-E2 in activating globin gene expression. Blood 1997, 89, 3925-3935.
6. Forsberg, E.C.; Downs, K.M.; Bresnick, E.H. Direct interaction of NF-E2 with hypersensitive site 2 of the beta-globin locus control region in living cells. Blood 2000, 96, 334-339.
7. Bean, T.L.; Ney, P.A. Multiple regions of p45 NF-E2 are required for beta-globin gene expression in erythroid cells. Nucleic Acids Res 1997, 25, 2509-2515, doi:10.1093/nar/25.12.2509.
8. Kaufmann, K.B.; Grunder, A.; Hadlich, T.; Wehrle, J.; Gothwal, M.; Bogeska, R.; Seeger, T.S.; Kayser, S.; Pham, K.B.; Jutzi, J.S.; et al. A novel murine model of myeloproliferative disorders generated by overexpression of the transcription factor NFE2. J Exp Med 2012, 209, 35-50, doi:10.1084/jem.20110540.
9. Jutzi, J.S.; Bogeska, R.; Nikoloski, G.; Schmid, C.A.; Seeger, T.S.; Stegelmann, F.; Schwemmers, S.; Grunder, A.; Peeken, J.C.; Gothwal, M.; et al. MPN patients harbor recurrent truncating mutations in transcription factor NF-E2. J Exp Med 2013, 210, 1003- 1019, doi:10.1084/jem.20120521.
10. Igarashi, K.; Kataoka, K.; Itoh, K.; Hayashi, N.; Nishizawa, M.; Yamamoto, M. Regulation of transcription by dimerization of erythroid factor NF-E2 p45 with small Maf proteins. Nature 1994, 367, 568-572, doi:10.1038/367568a0.
11. Clemens Bockelmann, L.; Basu, T.; Grunder, A.; Wang, W.; Breucker, J.; Kaiser, S.; Pichler, A.; Pahl, H.L. Concomitant constitutive LNK and NFE2 mutation with loss of sumoylation in a case of hereditary thrombocythemia. Haematologica 2021, 106, 1158-
1162, doi:10.3324/haematol.2020.246587.

12. Dunker, A.K.; Garner, E.; Guilliot, S.; Romero, P.; Albrecht, K.; Hart, J.; Obradovic, Z.; Kissinger, C.; Villafranca, J.E. Protein disorder and the evolution of molecular recognition: theory, predictions and observations. Pac Symp Biocomput 1998, 473-484.
13. Dunker, A.K.; Lawson, J.D.; Brown, C.J.; Williams, R.M.; Romero, P.; Oh, J.S.; Oldfield, C.J.; Campen, A.M.; Ratliff, C.M.; Hipps, K.W.; et al. Intrinsically disordered protein. J Mol Graph Model 2001, 19, 26-59, doi:10.1016/s1093-3263(00)00138-8.
14. Brodsky, S.; Jana, T.; Mittelman, K.; Chapal, M.; Kumar, D.K.; Carmi, M.; Barkai, N.  Intrinsically Disordered Regions Direct Transcription Factor In Vivo Binding Specificity. Mol Cell 2020, 79, 459-471 e454, doi:10.1016/j.molcel.2020.05.032.
15. Tokuriki, N.; Tawfik, D.S. Protein dynamism and evolvability. Science 2009, 324, 203- 207, doi:10.1126/science.1169375.
16. Chen, X.; Wen, S.; Fukuda, M.N.; Gavva, N.R.; Hsu, D.; Akama, T.O.; Yang-Feng, T.; Shen, C.K. Human ITCH is a coregulator of the hematopoietic transcription factor NFE2. Genomics 2001, 73, 238-241, doi:10.1006/geno.2001.6512.

Reviewer 2 Report

Comments and Suggestions for Authors

In the current manuscript, the authors analyzed NFE2, focusing on the role of the mutant form that is frequently identified in cases of myeloproliferative neoplasms. The results are clearly presented, and the manuscript is well summarized. Below are comments that may help further improve the manuscript.

  1. Although the authors state that mutant NFE2 (NFE2-226aa) protects wild-type NFE2 (NFE2-WT) from its degradation mechanism, additional explanation would help readers better understand this point. In particular, it is unclear how the authors determined which bands correspond to NFE2-WT in the Western blot shown in Figure 5A. Several NFE2 results in the figures appear as double bands, which may confuse readers.
  2. There appear to be differences in the localization of PLA signals among the various forms of NFE2, as shown in Figures 2C, 2F, and 2I. Because the authors note in the Introduction that NFE2 localization is associated with its function, it may be useful to discuss the localization of NFE2 with ICTH in more detail.
  3. The authors used β-actin as the loading control in most figures but used GAPDH in Figure 5A. Is there a reason for selecting different controls?

Author Response

Comment 1: Although the authors state that mutant NFE2 (NFE2-226aa) protects wild-type NFE2 (NFE2-WT) from its degradation mechanism, additional explanation would help readers better understand this point. In particular, it is unclear how the authors determined which bands correspond to NFE2-WT in the Western blot shown in Figure 5A. Several NFE2 results in the figures appear as double bands, which may confuse readers.

Response 1: We thank the reviewer for this valuable comment and apologize if some of the results were presented in a potentially confusing manner. It is a well-known phenomenon that wild-type NFE2 often appears as a double band in Western blots, depending on the cellular context. Several explanations have been proposed in the literature. The most widely accepted is that the presence of additional methionine residues allows translation to initiate at alternative start sites, generating a shorter open reading frame and a correspondingly smaller protein. [17-19] Other possibilities include phosphorylation of a subset of NFE2 molecules, which can cause a band shift, and partial proteolytic cleavage contributing to the lower band.

Importantly, the NFE2-226aa mutant can be clearly distinguished from NFE2-WT (Figure Reply Letter 1). To assist readers, we have updated the figure legends to indicate that NFE2-WT may appear as a double band. Furthermore, we have added the NFE2-226aa band to Figure 5A. We hope that these clarifications and modifications improve the presentation and make it easier for readers to distinguish wt NFE2 from the NFE2-226aa mutant.

Comment 2: There appear to be differences in the localization of PLA signals among the various forms of NFE2, as shown in Figures 2C, 2F, and 2I. Because the authors note in the Introduction that NFE2 localization is associated with its function, it may be useful to discuss the localization of NFE2 with ICTH in more detail.

Response 2: In order to address this reviewer’s comment, we have now included new data showing the localization of NFE2 and ITCH in more detail (see Fig. 7). To assess the subcellular localization of NFE2 and ITCH, we co-transfected GFP-tagged wt NFE2 or GFP-tagged NFE2-226aa together with either empty vector or mCherry-tagged ITCH into HEK-293T cells. SPY650 was
used for live-cell nuclear staining. GFP, mCherry, and SPY650 signals were detected by livecell fluorescence confocal microscopy.

Compared to wt NFE2, which is predominantly located in the nucleus, truncated NFE2-226aa showed a much weaker nuclear accumulation. Moreover, we show that ITCH relocalizes wt NFE2 from the nucleus to the cytoplasm, whereas it does not significantly affect the subcellular localization of NFE2-226aa.

Together, these new data support the hypothesis that ITCH reduces the transactivation activity of wt NFE2 by altering its subcellular localization from the nucleus to the cytoplasm, where it can be targeted for proteasomal degradation. NFE2-226aa, which retains the ability to interact with ITCH, is already enriched in the cytoplasm, where it acts as a decoy to prevent degradation of wt NFE2.

Comment 3: The authors used β-actin as the loading control in most figures but used GAPDH in Figure 5A. Is there a reason for selecting different controls?

Response 3: We thank the reviewer for this comment. There was no specific reason for using GAPDH instead of β-actin in Figure 5A. Both housekeeping proteins serve as equivalent loading controls.
